# Sulfate Decelerated Ferrous Ion-Activated Persulfate Oxidation of Azo Dye Reactive Brilliant Red: Influence Factors, Mechanisms, and Control Methods

**Chenliu Tang, Zhicheng Long, Yidan Wang, Dongze Ma and Xiaobiao Zhu ***

Department of Environmental Science and Engineering, College of Chemical Engineering, Beijing University of Chemical Technology, Beijing 100029, China
* Correspondence: zhuxiaobiao@mail.buct.edu.cn; Tel.: +86-10-64427356

**Abstract:** This study investigated the inhibition effects of sulfate on ferrous ion-activated persulfate oxidation of azo dye reactive brilliant red X-3B. The experimental results showed that the degradation efficiency of reactive brilliant red X-3B decreased from 100% to 63% in 60 min when the initial concentration of sulfate increased from 0 to 3 g/L. The ferrous/persulfate molar ratio had remarkable influence on persulfate oxidation capability in presence of sulfate. $SO_4^{2-}$ could coordinate with $Fe^{2+}$ and $Fe^{3+}$ in generating $FeSO_4$ ion pairs as well as $FeSO_4^+$ or $Fe(SO_4)_2^-$ complexes, which were difficult to activate persulfate and reduced the regeneration of $Fe^{2+}$. Radicals quenching and electron paramagnetic resonance experiments showed that $\cdot OH$ and $SO_4^{\cdot-}$ were responsible for the oxidation of reactive brilliant red X-3B; however, the addition of sulfate significantly inhibited the generation of $SO_4^{\cdot-}$, and then the generation of $\cdot OH$. Moreover, the corresponding Nernst equation indicated that high concentration of sulfate reduced the oxidation potential of $SO_4^{\cdot-}/SO_4^{2-}$. Experimental results proved that the adverse effects of sulfate on the persulfate oxidation could be counteracted either by batch addition of ferrous or by adding $Ba^{2+}$ to remove $SO_4^{2-}$ in the system.

**Keywords:** persulfate; sulfate; sulfate radicals; reactive brilliant red X-3B

## 1. Introduction

The refractory organic matters in wastewater are the key factors affecting the efficiency of wastewater treatment processes. Azo dyes, which are mostly aromatic compounds with complex structures and resistance to biodegradation, are widely used in the printing and dyeing industries [1,2]. Usually, advanced oxidation processes (AOPs) are used as pretreatment measures to degrade the molecular structures of the refractory organic matters in order to improve their biodegradability for further biological treatment [3,4]. Persulfate (PS) based AOPs employed sulfate radicals ($SO_4^{\cdot-}$) for oxidation of azo dyes in wastewater [5–7]. Compared with $\cdot OH$, $SO_4^{\cdot-}$ had higher redox potential, longer half-life time, and a broader application pH range, therefore, PS based AOPs had been used in degrading and mineralizing refractory pollutants [7–11]. At present, sulfate radicals are usually generated by PS or peroxymonosulfate (PMS) activation [12–14]. However, only a small amount of sulfate radicals could be produced by PS/PMS autolysis; therefore, the activation of PS/PMS was usually achieved by the addition of inexpensive, naturally abundant and environmentally friendly ferrous in the AOPs [15–18].

Other than organic pollutants, industrial wastewater contained a large number of co-existing inorganic ions, such as $SO_4^{2-}$, $Cl^-$, $NO_3^-$, $HCO_3^-$, $CO_3^{2-}$, $HPO_4^{2-}$, etc. Recently, the influence of these anions on the performance of the AOPs in degrading organic pollutants had attracted a lot of attentions. It had been shown that the presence of high concentration of $Cl^-$, $NO_3^-$, $HCO_3^-$, and $HPO_4^{2-}$ had a significant inhibition on the oxidation capacity of PS-based AOPs [19,20]. The inhibition was usually attributed to the quenching of active radical substances by inorganic ions, which could well explain the

effects of $Cl^-$, $HCO_3^-$, and $CO_3^{2-}$ ions on PS AOPs [21,22]. The high concentration of $Cl^-$ could react with the reactive radicals ($\cdot OH$ and $SO_4^{\cdot-}$) to produce a series of chlorine reactive radicals ($Cl\cdot$, $Cl_2^{\cdot-}$, $HOCl^{\cdot-}$) with lower oxidation potential, which reduced the concentration of free radicals with higher oxidation potential and inhibited the degradation efficiency of pollutants (Equations (1)–(3)) [23–25]. However, at low concentrations, $Cl^-$ could react with $\cdot OH$ or $SO_4^{\cdot-}$, which tended to annihilate themselves instead of degrading pollutants, producing chlorine reactive radicals with longer half-life time and slightly lower oxidation potential, and increased the total amount of reactive substances for pollutant degradation [23]. In addition, $HCO_3^-$ and $CO_3^{2-}$ could react with $SO_4^{\cdot-}$ to produce $HCO_3^{\cdot}$ and $CO_3^{\cdot-}$ radicals with much lower oxidation potential (Equations (4) and (5)), affecting the degradation performance of the AOPs system [26,27]. Other studies had shown that $PO_4^{3-}$ and $NO_3^-$ could also quench $SO_4^{\cdot-}$ to form $HPO_4^-$ and $NO_3^{\cdot}$ with lower oxidation potentials (Equations (6) and (7)) [16,28]. In addition, $HPO_4^{2-}$ could form complexes with $Fe^{3+}$, which inhibited the circulation between $Fe^{2+}$ and $Fe^{3+}$ during the advanced oxidation process, thus affecting the degradation ability of the system [29].

$$SO_4^{\cdot-} + Cl^- \rightarrow Cl\cdot + SO_4^{2-} \quad k = 2.7 \times 10^8 \ M^{-1}s^{-1} \tag{1}$$

$$Cl\cdot + H_2O \rightarrow HOCl\cdot + H^+ \quad k = 2.5 \times 10^5 \ s^{-1} \tag{2}$$

$$Cl\cdot + Cl^- \rightarrow Cl_2^{\cdot-} \quad k = 4.4 \times 10^8 \ M^{-1}s^{-1} \tag{3}$$

$$SO_4^{\cdot-} + HCO_3^- \rightarrow HCO_3^{\cdot} + SO_4^{2-} \quad k = 6.1 \times 10^6 \ M^{-1}s^{-1} \tag{4}$$

$$SO_4^{\cdot-} + CO_3^{2-} \rightarrow CO_3^{\cdot-} + SO_4^{2-} \quad k = 9.1 \times 10^6 \ M^{-1}s^{-1} \tag{5}$$

$$SO_4^{\cdot-} + HPO_4^{2-} \rightarrow HPO_4^- + SO_4^{2-} \quad k = 1.2 \times 10^6 \ M^{-1}s^{-1} \tag{6}$$

$$SO_4^{\cdot-} + NO_3^- \rightarrow NO_3^{\cdot} + SO_4^{2-} \quad k = 2.1 \ M^{-1}s^{-1} \tag{7}$$

The concentration of $SO_4^{2-}$ usually ranged from 1000 mg/L to 5000 mg/L in various industrial wastewaters [30], moreover, $SO_4^{2-}$ was a product from the activation of PS. Unlike $Cl^-$, $NO_3^-$, $HCO_3^-$, or $CO_3^{2-}$, $SO_4^{2-}$ could not react with $SO_4^{\cdot-}$ and $\cdot OH$ to produce secondary radicals. Although some studies had found that the presence of $SO_4^{2-}$ could inhibit the PS or PMS oxidation of organic pollutants, few studies have been conducted on the comprehensive influences of $SO_4^{2-}$ with other operation factors in PS oxidation system [31,32]. Furthermore, the influence mechanism of $SO_4^{2-}$ on PS oxidation was not thoroughly investigated, and only few studies referred to the inhibition of $SO_4^{2-}$ through the decreased removal efficiency of pollutants. The lack of profound studies on the negative effects of $SO_4^{2-}$ could not provide enough guidance for wastewater treatment. Therefore, it was necessary to clarify the performance and mechanism of $SO_4^{2-}$ affecting PS oxidation process.

In this study, $Fe^{2+}$ was employed to activate PS to generate radicals for the oxidation of reactive brilliant red (RBR) X-3B (Figure S1). The influence factors and mechanism of $SO_4^{2-}$ affecting PS oxidation efficiency was illustrated, and the feasible control methods for counteracting the adverse effects of $SO_4^{2-}$ was studied in order to provide support in the application of PS oxidation in wastewater treatment.

## 2. Results and Discussion

### 2.1. Effect of Different Anions

Inorganic ions, commonly contained in wastewater, would affect the degradation efficiency of pollutants by ferrous/PS system. In this section, the effects of three typical anions ($SO_4^{2-}$, $Cl^-$, and $NO_3^-$) on the performance of ferrous/PS degradation of RBR X-3B were investigated, and the experimental results were shown in Figure 1a.

At the concentration of 3 g/L, $SO_4^{2-}$ had the greatest inhibitory effect on the degradation of RBR X-3B, followed by $Cl^-$, while $NO_3^-$ had rare inhibitory effect. In the ferrous/PS system, the high concentration of $Cl^-$ would react with $SO_4^{\cdot-}$, reducing the concentration

of sulfate radical, thus inhibiting the degradation of RBR X-3B (Equations (1)–(3)). However, since this process generated a series of chlorine reactive radicals with lower oxidation potential but longer half-time, the overall degradation rate decreased within a narrower range, compared with the PS system without sulfate [23,33]. The secondary reaction rate between nitrate ions and sulfate radicals was lower than that of chlorine ions by 6–8 order of magnitudes (Equation (7)), thus the presence of nitrate had rare inhibition on the PS oxidation efficiency.

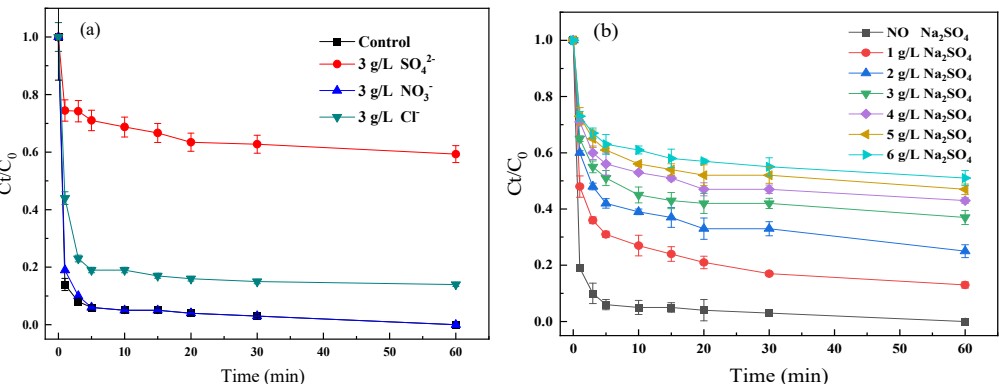

**Figure 1.** Effect of anions and sulfate on the degradation of RBR X-3B by ferrous/PS system: (**a**) anions; (**b**) sulfate. $[X-3B]_0 = 50$ mg/L, $[PS]_0 = 40$ mg/L, $Fe^{2+}$/PS molar ratio = 0.3, and pH = 7.

The effects of sulfate concentration on the ferrous/PS oxidation of RBR X-3B were shown in Figure 1b. The results showed that as sulfate concentration increased, the degradation efficiency of RBR X-3B decreased gradually. When the sodium sulfate dosage was 6 g/L, the degradation efficiency of RBR X-3B was reduced to less than 50% in the first 1 h of reaction, which indicated that the high concentration of $SO_4^{2-}$ would inhibit the degradation of pollutants in the PS system. Moreover, the system could only achieve about 40% of RBR X-3B removal efficiency under 6 g/L concentration of $SO_4^{2-}$. The degradation rate constant $k_{obs}$ decreased obviously with the increase of $SO_4^{2-}$ concentration, as shown in Figure S3a. When the concentration of $SO_4^{2-}$ was less than 3 g/L, the inhibition effect increased significantly with the increase of $SO_4^{2-}$ concentration. Compared to other methods to remove RBR X-3B in wastewater, for example, adsorption or electrical methods, the PS AOPs method could degrade more RBR X-3B in a shorter time and had no secondary pollutions [34,35].

### 2.2. Comprehensive Influences of $SO_4^{2-}$ with Other Operation Factors

In the ferrous/PS system, the concentration of PS directly affected the production of $SO_4^{\cdot-}$. Figure 2 showed that the degradation efficiency of RBR X-3B increased from 47% to 100% within 60 min when the PS concentration was increased from 10 mg/L to 40 mg/L. When PS concentration was more than 40 mg/L, further increase of the PS concentration contributed little to the increase of pollutants degradation efficiency. The results indicated that the increase of PS at low concentrations could increase the amount of $SO_4^{\cdot-}$, resulting in a rapid degradation of organic pollutants. Comparing Figure 2a with Figure 2b, the addition of $SO_4^{2-}$ comprehensively inhibited the degradation of RBR X-3B. When the PS concentration was 40 mg/L, the degradation rate reduced to 41.4% within 60 min, and the reaction rate constant $k_{obs}$ also decreased dramatically (Figure S3b), indicating that $SO_4^{2-}$ significantly inhibited the production of $SO_4^{\cdot-}$.

The effect of $Fe^{2+}$ dosage on the degradation efficiency of RBR X-3B was shown in Figure 2c,d. Without $Fe^{2+}$, the degradation efficiency of RBR X-3B was only 5%. When the $Fe^{2+}$/PS ratio increased to 0.3, the degradation of RBR X-3B increased rapidly to 96% within 30 min (Figure 2c). A small amount of $Fe^{2+}$ could activate PS to produce a large number

of sulfate radicals, but the excess $Fe^{2+}$ would quench the generated $SO_4^{\cdot-}$ (Equation (8)), which led to a decrease in pollutants degradation efficiency [36–38].

$$Fe^{2+} + SO_4^{\cdot-} \rightarrow Fe^{3+} + SO_4^{2-} \qquad (8)$$

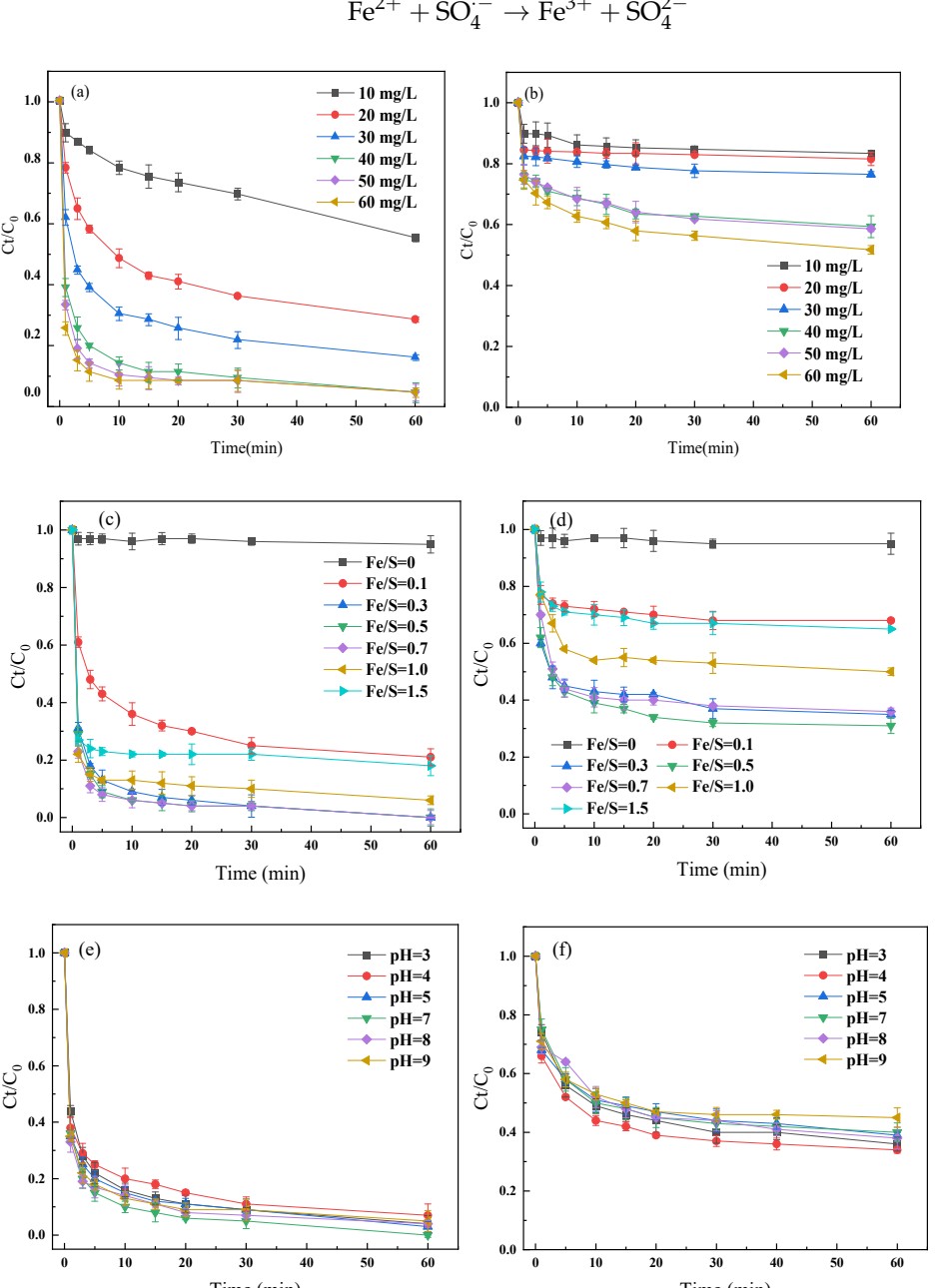

**Figure 2.** The degradation efficiency of RBR X-3B under different PS concentrations: (**a**) $SO_4^{2-}$ free and (**b**) with $SO_4^{2-}$; $Fe^{2+}$/PS ratios: (**c**) $SO_4^{2-}$ free and (**d**) with $SO_4^{2-}$, and initial pH: (**e**) $SO_4^{2-}$ free and (**f**) with $SO_4^{2-}$. $[Na_2SO_4]_0$ = 3 g/L, PS = 40 mg/L, $Fe^{2+}$/PS = 0.3, pH = 7, and $[X-3B]_0$ = 50 mg/L.

Figure 2d showed that the increase of $Fe^{2+}$ dosage could improve the degradation of RBR X-3B under high sulfate concentration. The highest reaction rate constant $k_{obs}$ without $Na_2SO_4$ was obtained at $Fe^{2+}$/PS = 0.3, but the $k_{obs}$ decreased significantly with the presence of $SO_4^{2-}$ (Figure S3b). However, $k_{obs}$ increased at first and then decreased because the excess $Fe^{2+}$ reacted with $SO_4^{\cdot-}$ and $\cdot$OH, which made $k_{obs}$ decrease dramatically. Under weak acidic conditions, $SO_4^{2-}$ could generate ion pairs ($FeSO_4$), which were difficult to activate PS, or generate a mixture of $FeSO_4^+$ and $Fe(SO_4)_2^-$ complexes, which reduced the regeneration of $Fe^{2+}$ from $Fe^{3+}$ [32,39,40]. When the $Fe^{2+}$/PS ratio was less than 0.1, $Fe^{2+}$

was insufficient to activate PS. However, when the $Fe^{2+}$/PS ratio was more than 1.5, the excess $Fe^{2+}$ not only formed $FeSO_4$ ion pairs, which made this part of $Fe^{2+}$ less reactive with $S_2O_8^{2-}$, but also quenched the generated $SO_4^{\cdot-}$ in the system (Equation (8)), resulting in a significant decrease of X-3B removal efficiency after the addition of $SO_4^{2-}$.

The pH of wastewater would affect the existing forms and concentrations of $Fe^{2+}$ and $Fe^{3+}$, since $Fe^{2+}$ and $Fe^{3+}$ would precipitate hydrolytically at pH 7 and 3.5, respectively, affecting the production of $SO_4^{\cdot-}$ in the ferrous/PS system [41]. $Fe^{2+}$ formed $Fe(H_2O)^{2+}$ at low pH, resulting in less $Fe^{2+}$ available for the activation of PS [42]. As shown in Figure 2e, the ferrous/PS system was able to degrade 90% RBR X-3B when the initial pH ranged from 3 to 9, but the degradation efficiency decreased significantly after the addition of $SO_4^{2-}$ (Figure 2f). The presence of $SO_4^{2-}$ not only affected the degradation efficiency in the fast reaction stage, but also had an adverse effect on the slow reaction stage, which was probably due to the fact that $SO_4^{2-}$ reduced the amount of $S{-}_4{}^-$ radicals in the ferrous/PS system.

*2.3. Influence Mechanism of $SO_4^{2-}$*

In order to investigate the mechanism of $SO_4^{2-}$ decelerating the ferrous/PS oxidation capacity, the $SO_4^{\cdot-}$ and $\cdot OH$ were quenched by ethanol (EtOH) and tert-butyl alcohol (TBA), respectively [38,43,44]. In Figure 3, the degradation efficiency of RBR X-3B within 60 min was 98% without the addition of EtOH or TBA. When TBA was added, the degradation efficiency of RBR X-3B decreased to 70%, indicating that $\cdot OH$ contributed 28% of the overall pollutant degradation. The degradation efficiency of RBR X-3B decreased to 32% after the addition of EtOH, and it could be presumed that $SO_4^{\cdot-}$ contributed 38% of the overall pollutant degradation. Previous studies had indicated that both $\cdot OH$ and $SO_4^{\cdot-}$ were generated in ferrous/PS system, synergistically degrading RBR X-3B. After the addition of $SO_4^{2-}$, the contributions of $SO_4^{\cdot-}$ and $\cdot OH$ to the overall pollutant degradation were reduced to 13.4% and 7.7%, respectively, indicating that $SO_4^{2-}$ inhibited the formation of $SO_4^{\cdot-}$ (Figure 3a,b). It could be assured that the decrease of $SO_4^{\cdot-}$ was due to less PS activation by the formed ion pairs ($FeSO_4$) from $Fe^{2+}$ and $SO_4^{2-}$. A decrease of $SO_4^{\cdot-}$ resulted in the decrease of $\cdot OH$ (Equations (9) and (10)) [19,33,45]. Meanwhile, as shown in Figure 3c, the addition of $SO_4^{2-}$ decreased the removal efficiency contributed by non-radical degradation from 32% to 19%. It was well known that in the ferrous/PS system, the removal of pollutants could also be achieved by the coagulation of $Fe^{3+}$ and the direct oxidation of $S_2O_8^{2-}$. The addition of $SO_4^{2-}$ generated complexes $FeSO_4^+$ and $Fe(SO_4)_2^-$ with $Fe^{3+}$, which led to a decrease of the coagulation process.

$$SO_4^{\cdot-} + H_2O \rightarrow SO_4^{2-} + \cdot OH + H^+ \tag{9}$$

$$SO_4^{\cdot-} + OH^- \rightarrow SO_4^{2-} + \cdot OH \tag{10}$$

EPR experiments were performed to further confirm the inhibition of the generation of free radicals by $SO_4^{2-}$ in the ferrous/PS system. As seen in Figure 4, the absorption spectrum of DMPO-$\cdot OH$ (marked by inverted triangles) indicated that $\cdot OH$ (hyperfine splitting constant: $a_H = 14.87$ G, and $a_N = 14.87$ G) was generated in the ferrous/PS system (Figure S2). There were also absorption signals of DMPO-$SO_4^{\cdot-}$ in the figure (diamond-shaped marker), indicating that $SO_4^{\cdot-}$ (hyperfine splitting constants: $a_H = 10.09$ G, $a_N = 13.94$ G, $a_{H\gamma 1} = 1.6$ G, and $a_{H\gamma 2} = 0.8$ G) was generated in the system. Most importantly, Figure 4b showed that the signal intensity of $SO_4^{\cdot-}$ and $\cdot OH$ became significantly weaker after $SO_4^{2-}$ was added, which further confirmed that $SO_4^{2-}$ could inhibit the generation of free radicals.

The presence of $SO_4^{2-}$ not only inhibited the production of $SO_4^{\cdot-}$, but also reduced the oxidation potential of the system. The $SO_4^{\cdot-}$ in the system was mainly generated through reaction as Equation (11), and the consumption of $SO_4^{\cdot-}$ could be simplified as Equation (12), then the corresponding Nernst equation was as Equation (13).

$$Fe^{2+} + S_2O_8^{2-} \rightarrow SO_4^{2-} + SO_4^{\cdot-} + Fe^{3+} \quad k = 17 \text{ M}^{-1}\text{s}^{-1} \tag{11}$$

$$SO_4^{\cdot-} + e^- \rightarrow SO_4^{2-} \tag{12}$$

$$E\left(SO_4^{\cdot-}\right) = E^\theta_{\left(\frac{SO_4^{\cdot-}}{SO_4^{2-}}\right)} + \frac{RT}{zF} \ln \frac{\left[SO_4^{\cdot-}\right]}{\left[SO_4^{2-}\right]} \tag{13}$$

where, $E^\theta_{(SO_4^{\cdot-}/SO_4^{2-})}$ was the standard half-reaction reduction potential;

R was the universal gas constant, $8.314472 \ J \cdot K^{-1} mol^{-1}$;

T was the absolute temperature;

F was the Faraday constant, $9.63845 \times 10^4 \ Cmol^{-1}$;

z was the number of electrons transferred in the half-reaction, which was 1 in this equation.

From Equation (13), the oxidation-reduction potential of $E(SO_4^{\cdot-}/SO_4^{2-})$ was influenced by the concentration of $SO_4^{2-}$, and the higher the concentration of $SO_4^{2-}$, the lower the $E(SO_4^{\cdot-}/SO_4^{2-})$. As seen in Figure S4, the $E(SO_4^{\cdot-}/SO_4^{2-})$ gradually decreased as the concentration of $SO_4^{2-}$ increased, and when the concentration of $SO_4^{2-}$ increased from 0 to 6 g/L, the oxidation-reduction potential of $E(SO_4^{\cdot-}/SO_4^{2-})$ decreased by 0.14 V, which was corresponded to the decreased degradation efficiency of RBR X-3B as sulfate concentration increased (Figure 1b). Figure S3 also indicated that the decreased reaction rate constant $k_{obs}$ after the addition of $SO_4^{2-}$ could also be explained that the presence of $SO_4^{2-}$ reduced the oxidation-reduction potential of $E(SO_4^{\cdot-}/SO_4^{2-})$, and therefore decreased the degradation efficiency of RBR X-3B.

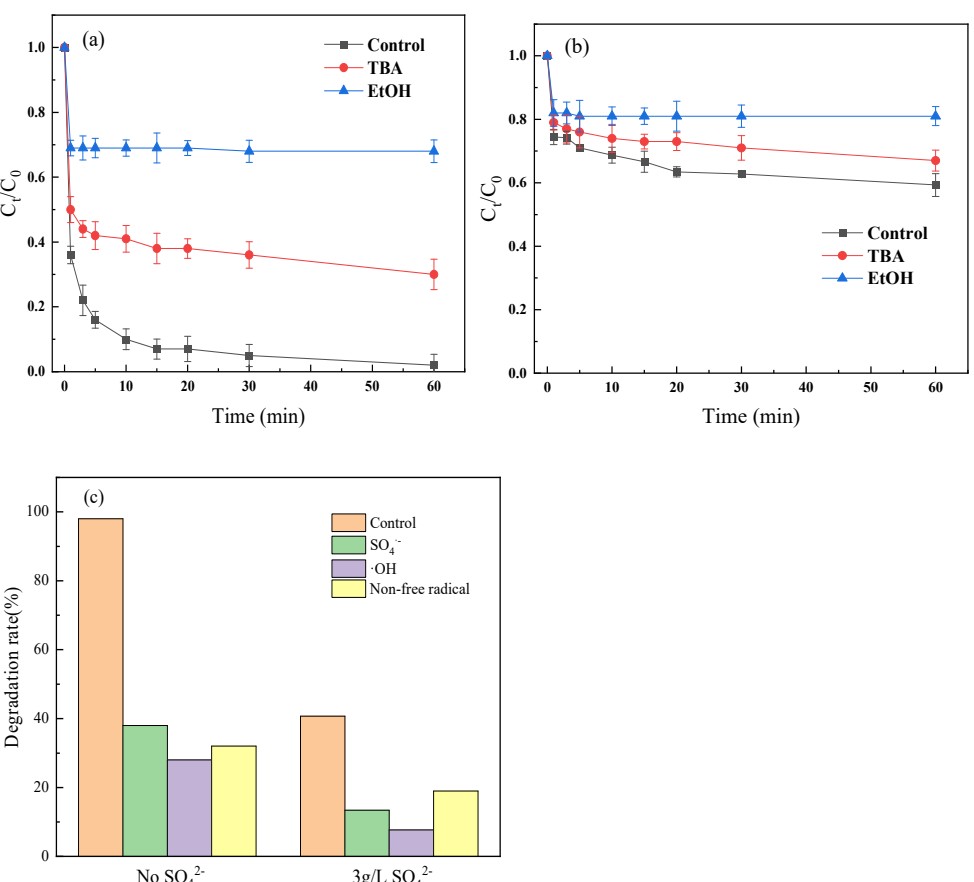

**Figure 3.** Free radical quenching experiment: (**a**) Na$_2$SO$_4$ free; (**b**) [Na$_2$SO$_4$] = 3 g/L; (**c**) contribution of reactive substances to the overall degradation efficiency. Reaction conditions: [PS] = 40 mg/L, Fe$^{2+}$/PS = 0.3, pH = 7, [TBA]$_0$ = 0.5 mM, and [EtOH]$_0$ = 0.5 mM.

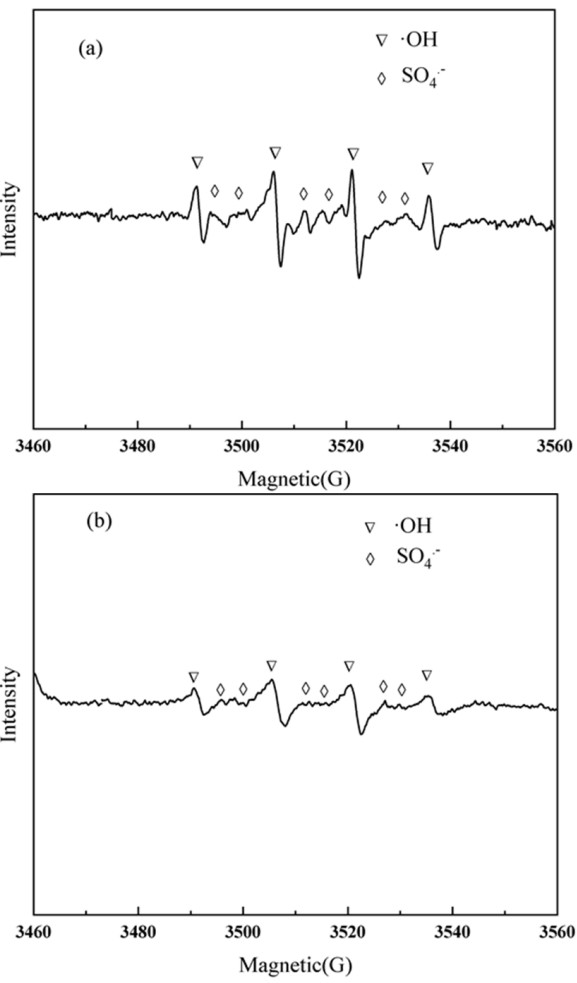

**Figure 4.** EPR absorption spectra of ferrous/PS system: (**a**) $Na_2SO_4$ free; (**b**) $[Na_2SO_4]_0$ = 3 g/L, $[PS]_0$ = 40 mg/L, $Fe^{2+}$/PS = 0.3, and pH = 7.

## 2.4. Methods to Counteract the Influence of $SO_4^{2-}$

The presence of $SO_4^{2-}$ greatly inhibited the degradation of pollutants in the ferrous/PS system. Thus, two methods were tested to increase the degradation efficiency of pollutants under high concentration of $SO_4^{2-}$: introducing barium to precipitate $SO_4^{2-}$ and batch addition of $Fe^{2+}$. The results were shown in Figure 5.

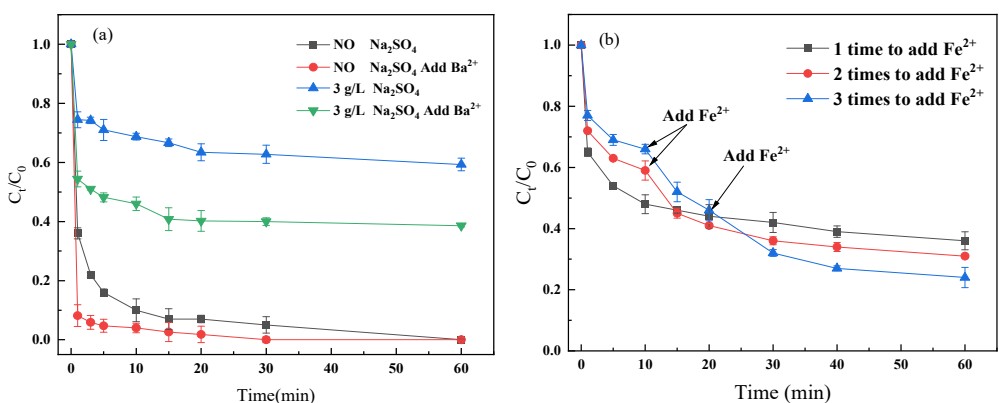

**Figure 5.** Methods to offset the adverse effects of $SO_4^{2-}$ on degradation of pollutants in ferrous/PS system: (**a**) $Ba^{2+}$ addition; (**b**) batch addition of $Fe^{2+}$. $[PDS]_0$ = 40 mg/L, $Fe^{2+}$/PS = 0.3.

In Figure 5a, when there was no $SO_4^{2-}$ in the system, the degradation efficiency reached 100% within 60 min, but when barium ions were added to the system, the RBR X-3B could be completely removed within 30 min, much faster than that without barium, indicating the $SO_4^{2-}$ generated from the PS activation could inhibit PS activation in turn. Then, $Ba^{2+}$ was added to the system with sodium sulfate concentration of 3 g/L, and the degradation efficiency of the pollutants increased from 40.7% to 61.4% within 60 min. Therefore, for actual wastewater treatment, a certain amount of $Ba^{2+}$ could be added to improve the degradation capacity of the system.

In addition, $Fe^{2+}$ could also be added in batches to improve the degradation efficiency. The purpose of adding $Fe^{2+}$ in batches was to increase the effective $Fe^{2+}$ concentrations as high as possible in the system and improve the activation efficiency of PS. Figure 5b showed that the addition of $Fe^{2+}$ in two or three batches could increase the degradation efficiency by 5.0% and 13.6%, respectively, without increasing the PS concentration and the total added $Fe^{2+}$ amount. Therefore, the degradation of pollutants could be improved by this method in practice.

## 3. Materials and Methods

### 3.1. Chemicals

Sodium persulfate ($Na_2S_2O_8$), ferrous sulfate ($FeSO_4 \cdot 7H_2O$), and sodium hydroxide (NaOH) were purchased from Tianjin Fuchen Chemical Reagent Factory (Tianjin, China). Sodium sulfate ($Na_2SO_4$), sulfuric acid ($H_2SO_4$), and ethanol ($C_2H_6O$) were purchased from Beijing Chemical Works. The free radical trapping agent DMPO ($C_6H_{11}NO$) was purchased from Aladdin Bio-Chem Technology Company (Shanghai, China), and RBR X-3B was purchased from a dye factory. All chemicals were of analytical purity, and the deionized water was prepared by Molgeneral-205 equipment (Shanghai Moller Scientific Instruments), and the stock solutions of all chemicals were freshly prepared.

### 3.2. Degradation Experiment

All experiments were carried out in 250 mL conical flasks. The flasks contained 100 mL of aqueous solution of RBR X-3B at certain concentrations. The solution was mixed by a magnetic stirrer and the stirring speed was adjusted to 200 rpm to ensure that the reaction system was uniform. The pH was adjusted by sodium hydroxide and sulfuric acid, and $SO_4^{2-}$ concentration was adjusted by adding sodium sulfate. The reaction was started by adding PS and then ferrous ions. At 1, 3, 5, 10, 15, 20, 30 and 60 min during the reaction, 4 mL of water sample was taken by a syringe into a 5 mL centrifuge tube. Then, the sample was centrifuged at 10,000 rpm for 3 min, and finally the residual absorbance of the sample was measured to calculate the degradation efficiency. All experiments were performed in three parallel.

### 3.3. EPR Experiment

The free radicals of the reaction system were measured by electron paramagnetic resonance spectrometry (EPR). The oxidation experiment was conducted in 250 mL conical flasks, and 2 mL of water samples was collected at specific time intervals and mixed with 2 mL of DMPO solution with a concentration of 100 mmol/L thoroughly. DMPO reacted with the free radicals to generate relatively stable spin radical adducts. Then, the adducts were measured using an electron paramagnetic resonance spectrometer (EPR) (parameter settings: central field: 3220 G, sweep width: 100 G, scan time: 60 s, magnification: 10.00 × 100, modulation width: 1 G, time constant: 0.1 s, power: 0.99800 mW), and finally the obtained EPR absorption spectra were analyzed.

### 3.4. Controlling Method

$Ba^{2+}$ inhibition experiment: all experiments were carried out in 250 mL conical flasks with 100 mL of solution of RBR X-3B at a certain concentration. The experiment was carried out under optimal reaction conditions by adding sodium sulfate to adjust the $SO_4^{2-}$

concentration and the PS and $Fe^{2+}$. Excess barium nitrate was added into the PS system to precipitate $SO_4^{2-}$ before the oxidation was started. Then, water samples were collected and measured to calculate the degradation efficiency. All experiments were performed in three parallel.

Batch dosing experiment: the initial concentration of sodium sulfate in the experimental system was set at 3 g/L and the PS concentration was 40 mg/L. A total of 14 mg $Fe^{2+}$ was added into the system at 10 min for one time addition, and at 10 min and 20 min for two times addition, after the oxidation was started. Then, all experiments were performed as above.

## 4. Conclusions

In this study, the effects of high sulfate ions on the persulfate-based advanced oxidation were systematically analyzed in the ferrous/PS system by investigating the degradation of RBR X-3B under different PS concentration, $Fe^{2+}$/PS ratio, and initial pH. It was found that high $SO_4^{2-}$ concentration would decrease the total oxidation efficiency of the system by inhibiting the activation of persulfate. The degradation efficiency of pollutants decreased with the increase of the concentration of the sulfate ions. According to free radical quenching experiments and EPR detection, it was confirmed that $SO_4^{2-}$ could reduce the generation of $SO_4^{\cdot-}$, and then the total amount of radicals in the system. On the basis of illustrating the mechanism of $SO_4^{2-}$ inhibition, the addition of $Ba^{2+}$ and batch addition of $Fe^{2+}$ were proposed to improve the oxidation capacity of the system at high $SO_4^{2-}$ concentration.

**Supplementary Materials:** The following supporting information can be downloaded at: https://www.mdpi.com/article/10.3390/catal12101207/s1, Figure S1: Molecular structure formula of reactive brilliant red X-3B; Figure S2: Conversion of DMPO-SO$_4$ to DMPO-OH; Figure S3: Variation of the chemical reaction rate constant $k_{obs}$ for the system under different reaction conditions: (a) $Na_2SO_4$ concentration, (b) PS concentration, (c) pH, (d) $Fe^{2+}$/PDS. Figure S4. Variation of $[E(SO_4^{\cdot-}/SO_4^{2-}) - E\theta(SO_4^{\cdot-}/SO_4^{2-})]$ of the system at different $Na_2SO_4$ concen-trations.

**Author Contributions:** Conceptualization, X.Z.; Investigation, Z.L.; Methodology, C.T.; Validation, Y.W. and D.M.; Writing—original draft, Z.L., Y.W. and X.Z. All authors have read and agreed to the published version of the manuscript.

**Funding:** This research was funded by the Fundamental Research Funds for Central Universities, grant number ZY2202.

**Data Availability Statement:** Not applicable.

**Conflicts of Interest:** The authors declare no conflict of interest.

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
