# Peer review of "Sulfate Decelerated Ferrous Ion-Activated Persulfate Oxidation of Azo Dye Reactive Brilliant Red: Influence Factors, Mechanisms, and Control Methods"

_catalysts, doi:10.3390/catal12101207_

Round 1

Reviewer 1 Report

This manuscript assessed the effect of sulfate on the degradation of azo dyes and elucidated the mechanism. The content of the study was substantial and the figures were well made. I thought a few minor changes were needed:

1. Line 59-60, the author introduced the concentration of SO42- in actual industrial wastewater, but lacked literature support. In addition, the concentrations in the authors' study appeared to be far above this level and would like to provide an explanation.

2. The equations (9) and (10) seemed to show that pH had a great influence on SO42-, while figure 2 showed no significant difference. The author should give an explanation.

3. Abstract and keywords should be further improved.

Author Response

Response to comments from Reviewer 1

This manuscript assessed the effect of sulfate on the degradation of azo dyes and elucidated the mechanism. The content of the study was substantial and the figures were well made. I thought a few minor changes were needed:

Comments: 1. Line 59-60, the author introduced the concentration of SO42- in actual industrial wastewater, but lacked literature support. In addition, the concentrations in the authors' study appeared to be far above this level and would like to provide an explanation.

Response: Thanks for your question. The literature has been added. Here, the concentration of SO42- ranged from 1000 mg/L to 5000 mg/L. The following study used the concentration 1-6 g/L Na2SO4. It was at nearly the same level.

Comments: 2. The equations (9) and (10) seemed to show that pH had a great influence on SO42-, while figure 2 showed no significant difference. The author should give an explanation.

Response: The equations (9) and (10) reflected how one SO4·- generated one ·OH in the system, and also indicated that the total amount of radicals (SO4·- plus ·OH) was stable. The pH values would affect their proportion. Since both SO4·-and ·OH could degrade pollutants, the results in Figure 2 showed no significant difference in the removal efficiency.

Comments: 3. Abstract and keywords should be further improved.

Response: Thanks. The Abstract and keywords had been improved.

Reviewer 2 Report

This paper nicely presented the effect of sulfate ions in Fe2+/PS. They performed the experiment well. The presented results are explained very well and can be accepted after minor revision

  1. Please refer to the following articles related to the present study

https://doi.org/10.1016/j.ultsonch.2018.07.009

https://doi.org/10.1016/j.cej.2021.133002

  1. Please correct the superscript in the rate constant
  2. Line 60-62; What is the specialty of sulfate ions compared to other ions? What is the rate constant of sulfate radical with sulfate ions?
  3. What is the source of Fe2+ in the Fe2+/PS system? If it is Fe2SO4, is there any effect from sulfate ions?
  4. Figure 2 seems to be complex. It is better to divide the Figure
  5. Figure 4, it is better to emphasize on the line spectra of the radicals only
  6. As supporting data, is it possible to measure the Fe3+ formation during oxidation?

Author Response

Response to comments from Reviewer 2

This paper nicely presented the effect of sulfate ions in Fe2+/PS. They performed the experiment well. The presented results are explained very well and can be accepted after minor revision

Comments: 1. Please refer to the following articles related to the present study

https://doi.org/10.1016/j.ultsonch.2018.07.009

https://doi.org/10.1016/j.cej.2021.133002

Response: The above articles had been referred.

Comments: 2. Please correct the superscript in the rate constant 

Response: Thanks. All the errors in the rate constants had been corrected.

Comments: 3. Line 60-62; What is the specialty of sulfate ions compared to other ions? What is the rate constant of sulfate radical with sulfate ions?

Response: Sulfate ions could not react with sulfate radical while other ions could, as shown in equations (1-7). Therefore, there was no rate constant of sulfate radical with sulfate ions.

Comments: 4. What is the source of Fe2+ in the Fe2+/PS system? If it is FeSO4, is there any effect from sulfate ions?

Response: Thanks. The source of Fe2+ is FeSO4. But the concentration of the added FeSO4, about was very low (about 25-65 mg/L) compared to the studied level of sulfate ions (1-6 g/L). Therefore, the effect could be ignored.

Comments: 5. Figure 2 seems to be complex. It is better to divide the Figure

Response: We appreciate your suggestion. But here we intended to study the comprehensive influences of PS concentration, Fe2+/PS ratios, initial pH with or without sulfate. In order to conveniently compare the degradation efficiency under different factors, all the results were present together.

Comments: 6. Figure 4, it is better to emphasize on the line spectra of the radicals only

Response: Thanks for the suggestion. We have redrawn the Figure and only radicals were emphasized.

Comments: 7. As supporting data, is it possible to measure the Fe3+ formation during oxidation?

Response: Yes, it is possible to measure the concentration of Fe3+. Here, we did not do this because this article emphasized the effect of sulfate on the PS oxidation efficiency. The generation of Fe3+ came from Fe2+ catalyzing PS and could provide information about the ferrous utilization. It was a good suggestion, and we will take it into consideration in future studies.

Reviewer 3 Report

Sulfate decelerated ferrous ion-activated persulfate oxidation 2 of azo dye reactive brilliant red: influence factors, mechanisms, 3 and control methods

I think it is a very well written work and full of relevant new information, I recommend its publication taking into account these remarks.

·        In the citing part of advanced oxidation processes it is better to cite reviews instead of case study articles, for this I propose the following new reviews:

1.Elkacmi, R., & Bennajah, M. (2019). Advanced oxidation technologies for the treatment and detoxification of olive mill wastewater: a general review. Journal of Water Reuse and Desalination9(4), 463-505.

2.Wang, J., & Zhuan, R. (2020). Degradation of antibiotics by advanced oxidation processes: An overview. Science of the Total Environment701, 135023.

3.Ma, D., Yi, H., Lai, C., Liu, X., Huo, X., An, Z., ... & Yang, L. (2021). Critical review of advanced oxidation processes in organic wastewater treatment. Chemosphere275, 130104.

·        It was necessary to add some references on the works applied for the degradation of reactive brilliant red (RBR) X-3B, by adding the disadvantages of these techniques compared to the proposed process.

·        Line 59: Variation in SO42-  concentration is according to what exactly and please add references.

·        Have you not tested the optimal parameters found to monitor the degradation of RBR X-3B.

·        Why the choice was put on 3 g/l for the 3 anions.

·        The addition of Ba2+ and the addition of Fe2+ to improve the oxidation capacity will not influence the technical and economic feasibility of the treatment

·        The conclusion must be improved by results.

Author Response

Response to comments from Reviewer 3

I think it is a very well written work and full of relevant new information, I recommend its publication taking into account these remarks.

Comments: In the citing part of advanced oxidation processes it is better to cite reviews instead of case study articles, for this I propose the following new reviews:

1.Elkacmi, R., & Bennajah, M. (2019). Advanced oxidation technologies for the treatment and detoxification of olive mill wastewater: a general review. Journal of Water Reuse and Desalination9(4), 463-505.

2.Wang, J., & Zhuan, R. (2020). Degradation of antibiotics by advanced oxidation processes: An overview. Science of the Total Environment701, 135023.

3.Ma, D., Yi, H., Lai, C., Liu, X., Huo, X., An, Z., ... & Yang, L. (2021). Critical review of advanced oxidation processes in organic wastewater treatment. Chemosphere275, 130104.

Response: The author had read the above articles detailly, and the related articles had been referred.

Comments: It was necessary to add some references on the works applied for the degradation of reactive brilliant red (RBR) X-3B, by adding the disadvantages of these techniques compared to the proposed process.

Response: Thanks for the suggestions. We have added references and compared with other methods that treated RBR X-3B on lines 103-106, as “Compared to other methods to remove RBR X-3B in wastewater, for example, adsorption or electrical methods, the PS AOPs method could degrade more RBR X-3B in a shorter time and had no secondary pollutions.”

Comments: Line 59: Variation in SO42- concentration is according to what exactly and please add references.

Response: Thanks for suggestions. The reference has been added and this concentration level was common in high salinity wastewater.

Comments: Have you not tested the optimal parameters found to monitor the degradation of RBR X-3B.

Response: We did not designed experiments to obtain the optimal parameters for RBR X-3B degradation, but the key parameters that could affect the RBR X-3B degradation, including PS concentration, Fe2+/PS ratios and initial pH, were thoroughly investigated in Figure 2. Moreover, the main purpose of this study was to reveal the effects of sulfate ions, and the results would also be suitable in optimal conditions.

Comments: Why the choice was put on 3 g/l for the 3 anions.

Response: In some industrial wastewater, the total salinity would be as high as 5-10% (w/w). The common anions in high-salinity wastewater were SO42-, NO3- and Cl-. The concentration of 3 g/L referred to these wastewaters, and the same concentration made it easy to compare the effect on pollutants degradation of each other.

Comments: The addition of Ba2+ and the addition of Fe2+ to improve the oxidation capacity will not influence the technical and economic feasibility of the treatment

Response: It will not, because the batch addition of Fe2+ could be carried out easily in practice. And the addition of Ba2+ would be the second choice when there was wastewater containing Ba2+ could be used.

Comments: The conclusion must be improved by results.

Response: Thanks for your suggestions. The conclusion has been improved thoroughly and some sentences have been rewritten. It would be better now.